# Research on Influence of Tool Deformation in the Direction of Cutting and Feeding on the Stabilization of Vibration Activity during Metal Processing Using Metal-Cutting Machines

**DOI:** 10.3390/s23177482

**Published:** 2023-08-28

**Authors:** Lapshin Viktor, Turkin Ilia, Dudinov Ilia

**Affiliations:** Department of Automation of Production Processes, Don State Technical University, 344000 Rostov-on-Don, Russia; tur805@mail.ru (T.I.); ilya.sandman@yandex.ru (D.I.)

**Keywords:** cutting system, cutting dynamics, regenerative effect, vibrations

## Abstract

The article is devoted to the issues regarding assessment of the impact of changes in the delay in the feed direction on the dynamics of metal processing with metal-cutting machines. Here, for the first time, it is proposed that we take into account, when constructing models of the cutting control system, the real value of the delay value. It is this model that allows us to adequately consider the dynamics of the cutting process, by clarifying the effect of vibration regeneration. In this article, much attention is paid to describing the development of a measuring system that allows the calculation of the real value of the feed during cutting. It describes a series of experiments, and shows the results of data processing, using software developed by the authors. The studies conducted have shown that, in addition to the vibrations of the cutting tool in the feed direction, the vibration activity of the tool in the cutting direction plays an essential role in ensuring the regenerative effect. It should be noted that, currently, this is a new area of knowledge regarding processes in metalworking, and it requires further study.

## 1. Introduction

Since the middle of the 20th century, one of the main reasons for the loss of stability of cutting control systems has been the problem of the system self-excitation when cutting along the track, which has been called the regenerative effect [1,2,3]. In the Soviet scientific school of the 1960s, this problem was also considered [4]; however, it was considered to be not the main cause, but one of many possible ones. Through the further development of scientific thought in this direction, the influence of the main nonlinearities on the dynamics of cutting processes, taking into account the regeneration of vibrations, was investigated [5,6]. It is worth noting here that these studies [1,2,3,5,6] consider the scalar case of modeling a system of vibrational movements of the tool tip. The modern understanding of the causes of cutting control systems’ self-excitation in metal-cutting machines is largely based on the research of previous authors; however, the models used in this case are somewhat more complicated, so, in [7], a variant of the model of cutting system dynamics, with two degrees of freedom, is considered. In [8], the effect of vibration regeneration on the milling process is considered, but the model is scalar in nature. In [9], a scalar model for the case of orthogonal cutting is also presented; here, the author reveals the formation of periodic, quasi-periodic, and chaotic movements of the tool tip, the transition to which is caused by a regenerative effect. In [10], the author’s method of suppressing vibrations during cutting caused by the regenerative effect is presented. An interesting work in this field is [11], an article in the list of sources, which discusses the simulation of vibration regeneration during turning, through a partial differential deformation model. Of interest is also the work [12], for which the authors developed bifurcation diagrams defining the relationship between the applied processing parameters and the amplitude of the resulting limit cycles. One of the most famous modern researchers on the issues of cutting process dynamics, as well as how they are affected by the regeneration of vibrations when cutting along the track, is Stépán G [13], whose work examines the effect of the delay constant on the nonlinear dynamics of the cutting process. In addition to theoretical research in the field of the regeneration of tool tip vibrations during cutting, a fairly large number of practically orientated works in this area are known of. Thus, in [14], the development and implementation of a subsystem for regulating the cutting speed in numerically controlled (CNC) machines is considered, and allows the avoidance of cutting modes that cause the subsystem self-excitation of deformational tool movements. A similar work is [15], in which the issue of optimizing the cutting system according to the criterion of minimizing the vibrations of the tool tip is considered. In the works [16,17,18], methods for the analytical prediction of vibration regeneration are considered, the use of which, in cutting control systems, will eliminate the variants of stability loss in the subsystem of deformation movements.

One of the factors uniting all these works [1,2,3,4,5,6,7,8,9,10,11,12,13,14,15,16,17,18] is the presence in the mathematical model of a time constant that characterizes “time delay” signals in this model. The issues of stability assessment for models of systems with a delay are a separate, extremely interesting, mathematical problem [19]. However, in all these cases, the time delay was introduced into the mathematical model as a simulation parameter. In the case of a real metalworking process with metal-cutting machines, such a time delay depends on the rate of change in the coordinates of the system state of deformation movements of the tool tip. Mathematical models do not currently exist that take into account such a dependence of the delay value on the speed coordinates of the cutting process state.

Of interest are works in which mathematical models of dynamics were considered in variations relative to a constant equilibrium point, considered in a moving coordinate system, the movement of which is determined by the trajectories of the machine’s executive elements [20,21]. The trajectories of the machine’s actuating elements are considered during longitudinal turning, with constant values of the workpiece revolution speeds, and longitudinal feeding, at zero speed of the transverse caliper. Therefore, when considering the regenerative effect, it is assumed that the value of the lagging argument is constant. When forming the feed, two arrays of trajectories are considered, which are shifted relative to each other in time via a constant value of the lagging argument. As a result, the effects of synchronization and asynchronous interactions, as well as the effects of complex formation, attracting sets of feeds, and corresponding cutting forces, are observed, which can lead to the formation of chaotic attractors of deformations [22,23,24,25]. In addition to the regeneration of vibrations during cutting, the dynamics of deformation movements of the tool tip are also significantly affected by the process of temperature formation in the contact zone of the tool and the workpiece, which includes the conversion of irreversible transformation power into heat generated during cutting [26].

The declared approaches to the formation of mathematical models of cutting control systems [20,21,22,23,24,25,26] are largely based on the representation of forces that hinder the formative movements of the tool in the coordinates of deformations of the tool tip. With this representation, the lag constant, in the real cutting process determined by the spindle revolution speed and the deformation mixing of the tool tire in the cutting direction, should also be represented in the same coordinates. In fact, the delay constant “time delay” is determined by the speed of revolution of, and the rate of deformation displacements of, the tool tip, in the direction of revolution. Taking into account this previously unexplored fact significantly changes the model of the cutting system, which can no longer be scalar.

At the same time, modern measuring systems continue to develop, including measuring systems built on eddy current sensors [27,28]. For example, in [28], the authors propose the measurement of the transverse vibrations of a disk cutter when cutting wood with a wood-processing machine. The complexity of the application of the method of developing measuring systems in the case of the vibration control of the cutting tool in meta-processing are mechanical limitations (the presence of metal chips, the use of cooling systems for the cutting zone, etc.), and the fact that a wide range of radiated energy, including electromagnetic, is released during the cutting process. As a result, this method of measuring the vibrations of the cutting tool becomes difficult and unreliable. In addition to eddy current methods for measuring vibrations, methods based on the evaluation of the power of the vibroacoustic signal have become widespread in metalworking machines [29]. However, to control the degree of wear of the cutting tool, it is impossible to evaluate the vibration of the top of the cutting tool via this signal, due to the strong noise level of the signal. One of the most modern measurement methods is the method of 3D laser Doppler scanning vibrometry [30]. The use of this method directly when cutting metals is limited by the factors that we voiced for the case of eddy current sensors (the presence of metal chips, the use of cooling systems for the cutting zone, etc.). The most reliable method of measuring vibrations of the cutting tool is a method based on the widespread use of piezoaccelerometers [31]. In the case of a protective casing for these devices, they can be used almost in the cutting zone; in this regard, the captured signal will be the most informative for subsequent analysis.

All of this informs the purpose of this study, which consists of assessing the dynamics of the cutting system according to a mathematical model with a lagging argument, the value of which depends on the coordinates of the state of the system. The cutting forces, in this case, should include a subsystem of the force reaction formed on the back faces of the cutting wedge of the tool [32,33], which allows the taking into account of the power of irreversible transformations allocated at this contact [26]. In addition, the influence of the sets of deformation displacements formed in the phase space, which affect the super-low-frequency reactions of the cutting system, is subject to further research.

Summarizing the above, and using the terminology of automatic control theory, we note that deformations in the directions of the longitudinal and transverse feeds cause a decrease in the area of the cut layer; that is, they form a negative feedback. Deformations in the direction of the cutting speed lead to flexible positive feedback, which cannot but affect the properties of the dynamic cutting system. This joint is flexible, because it changes forces depending on the strain rate. This also changes the lag argument. The discovery of this connection affecting the properties of the cutting system, discussed in the article, complements the knowledge about it, which determines the scientific novelty and significance of the studies listed below.

## 2. Development of a Basic Mathematical Model

To clarify the mathematical model, we will consider the main axes of cutting tool deformation, as well as the orientation of the force reaction to the shaping movements of the tool, which are shown in Figure 1. 

Taking into account the scheme of the force reaction decomposition from the cutting process to the shaping movements of the tool presented in Figure 1, the forces decomposed along the axes of deformation can be represented as follows [34,35]:(1)$$\left\{\begin{array}{c} F_{f}=\chi_{1}F\\ F_{p}=\chi_{2}F\\ F_{c}=\chi_{3}F \end{array}\right.$$
where *χ*_1_, *χ*_2_, *χ*_3_ are the coefficients of decomposition of the cutting force *F* on the axis of deformation of the tool. The *F_c_* component has the greatest value here, which most accurately reflects the cutting force itself, and determines the vibration activity of the tool in the direction of the *z* axis. The force itself, which prevents the shaping movements of the tool, can be represented as the product of the cut layer area by some indicator characterizing the chip pressure on the front face of the cutting wedge of the tool, as presented in the expression below [20,21,22,23,24,25]:(2)$$F=\rho t_{p}S$$
where *ρ* is the constant determining the value of the specific chip pressure, per millimeter of the area of the layer cut during cutting, *t_p_* is the depth of the cut layer (mm), and *S* is the feed per revolution (mm). It should be noted here that the oscillation of the cutting force may be associated not only with the mechanism of self-excitation during cutting, but also with the complex molecular structure of the processed metal [36]. However, in the present paper, we do not consider this mechanism of oscillation generation.

It should be noted here that the actual cutting depth and the actual feed per revolution will depend on the vibration values of the tool tip; in the case of the cutting depth, it will be determined as follows (see Figure 1):(3)$$t_{p}=t_{p0}-y$$
where *y* is the amount of deformation of the tool tip in the radial direction.

As for the amount of feed per revolution, this will be determined as the solution of the following integral [20,21,22,23,24,25]:(4)$$S=\int_{t-T}^{t}\left(V_{f}-\frac{dx}{dt}\right)dt$$
where *V_f_* is the speed in the feed direction, *T* is the lag time constant calculated as the revolution time of the spindle with the part fixed in it, and *dx/dt* is the speed of the vibrations of the cutting tool tip in the feed direction.
(5)$$S=V_{f}T-x\left(t\right)+x\left(t-T\right)$$

In Expression (5), *T*—time delay depends on the magnitude of the oscillations of the cutting wedge tip in the direction of the *z* axis (see Figure 1). We illustrate this explanation with the following figure (see Figure 2).

In Figure 2, two variants of the growth of the cutting path are considered; the red color indicates the option in which cutting is carried out at the speed of *Vc* = 2100 mm/s, and without taking into account the vibrations of the cutting wedge in the direction of the *z* axis; in black is the same variant of the cutting path, but with vibrations superimposed on it in the direction of the *z* axis. For the convenience of reasoning, we assumed that the diameter of the processed product (workpiece) is assumed to be 32 mm, which gives approximately 100 mm of path per revolution of the spindle. As can be seen from the presented figure, under the conditions of the vibration activity of the cutting wedge, the actual cutting period (the delay value in Equation (5)) differs significantly from the value determined via the cutting program.

Based on these considerations, the real feed defined by Expression (5) will differ from the given feed, and not only will this difference consist of the two last terms of the sum, but the first term of this sum will also differ, due to the variation in the time constant T (see Figure 2).

Thus, the regenerative effect during cutting will be determined not only by the ratio of the trace during cutting and the current vibrations of the tool in the feed direction, but also by the vibrations of the cutting wedge in the cutting direction. All of this together creates a complex system of the related vibration activity of the cutting tool, which will determine the vibration levels during cutting.

## 3. Description of the Experiment and Data Processing Methodology

To determine the feed value during cutting, based on our previous reasoning, it will be necessary to evaluate the vibrational activity of the tool tip, both in the feed direction (along the *x* axis) and in the cutting direction (along the *z* axis). Thus, the vibration activity registration system should include two vibration transducers mounted on the corresponding axes of the instrument. To solve this problem, we have developed a measuring system that includes two vibration measurement sensors. The upgraded 1K625 universal lathe, shown in Figure 3a, was used as a test bench (for the possibility of smoothly changing the cutting speed). The parameters of the mode in the experiments were as follows: feed 0.11 mm/rev, cutting depth *t_p_*_0_ = 1 mm. Over the entire period of the experiment, these cutting parameters remained unchanged. The cutting speed varied in increments of 167 mm/s (10 m/min), from a value of 1167 mm/s (70 m/min) to a value of 2833 mm/s (170 m/min). As a tool, we used the holder MR TNR 2525 M22, and a pentahedral plate 10113-110408 T15K6 as a cutter on it (see Figure 3b), with the angle in the upper part (angle of attack) as 350, and the main angle in the plan as 800 (the angle between the projection of the main cutting edge on the main plane and the feed direction); both the holder and the plate are manufactured in Russia. In the experiment, a shaft made of steel grade 45 (GOST) was processed; this steel grade is widely used not only in Chelyabinsk, Russia (GOST 2591-2006—Steel 45 (st45)—high-quality structural carbon) [37], but also in the USA and Germany; the American standard ASTM A568M marks this type of steel as AISI 1045 [38], and Germany’s DIN17200 standard is Ck45. Steel 45 (st45) contains: from 0.42 to 0.5% carbon (this can be traced in the name of the steel grade), and 97% iron, as well as percentages of silicon, manganese, nickel, sulfur, phosphorus, chromium, copper, and arsenic [39]. The shaft made of grade 45 steel was created using hot rolling technology; before the experiment, the shaft was cut to a length of 75 cm, and then a preliminary finishing treatment of the shaft surface, and precise alignment, were carried out. The choice of this steel grade is associated with its wide use in mechanical engineering, which is due to the high quality of structural steel, and its relatively low price on the steel market. In the measurement system shown in Figure 3, two sensors (vibration transducers) manufactured by GLOBAL TEST (manufacturing country: Sarov, Russia) AP2098-500, were connected to the amplifier converter of GLOBAL TEST AR13, which converts the current signal into voltage, and amplifies it. These vibration converters have an analog output signal with a very high natural cutoff frequency of 48 kHz; the cutting process itself has a base vibration frequency in the range from 1 to 4 kHz. To digitize such a signal, it is necessary to have a quantization frequency of at least 8 kHz; in this regard, an ADC of the company L-CARD (manufacturer country: Russia) E14-440 AD/DA CONVERTER, with a USB 2.0 cable (USB Type B), was used. The measurement system, shown in the figures, uses two sensors (vibration transducers) to create a GLOBAL TEST (the country of origin is Russia) AP2098-500 connected to a stable converter for the GLOBAL TEST AR13, which converts the current signal into voltage, and observes it.

The interface of the ADC program is a dialog box, in which we can determine the duration of data recording, and set the quantization frequency of the signal taken from the sensors. The program also outputs an array of stored vibration acceleration data in the form of two vectors, a vector of vibration acceleration values, and a vector of discreet numbers. As we used the value of the quantization frequency of the signal at 10 kHz, 10,000 values are represented in every vector, for every second of measurements. Such a high quantization frequency is due to the fact that the main events associated with vibrations taken from the instrument occur in the frequency range from 1 kHz to 5 kHz. According to the requirements of the Nyquist–Shannon theorem, to restore such a signal from a discrete value, a quantization frequency of at least twice the natural frequency of the original analog signal is required.

The values of vibration acceleration obtained in the experiments were processed by us in the MATLAB-14 mathematical modeling environment, where programs were written that allow the double integration of vibration accelerations, with filtering of the average value, calculated here via the moving average method. In addition, programs were compiled for calculating the RMS value of vibration accelerations (MCP) at the measurement sites, as well as a program for the spectral analysis of the received signals. The results of preprocessing one of the measured signals are shown in Figure 4.

As can be seen from Figure 4, the integration technique we have adopted, as a whole, adequately converts the vibration acceleration signal into a vibration velocity and vibration displacement signal, in the direction of the *z* axis. However, to verify the correctness of the obtained solution, at every step of the experiment, we carried out a comparative analysis of the signal spectra shown in Figure 4. The results of such a comparison are presented in Figure 5, in the form of the power spectra of the vibration acceleration signals, vibration velocity, and vibration displacement of the tool tip, in the direction of the *z* axis.

As can be seen from Figure 5, in all considered cases of the signal under study, there is a surge in vibration power at the frequency from 2000 to 3000 Hz; however, as the integration increases, the role of vibrations at low frequencies increases, which is explained by the inertia of the instrument, which does not have time to respond to the high-frequency harmonics of the vibration acceleration signal.

Thus, we conducted twelve experiments with different cutting speeds, the data from which we processed using our own development programs.

## 4. Results and Discussion

It is convenient to present the data obtained as a result of a series of experiments for further analysis in the form of MCP values (RMS values), which were calculated according to the following formula:(6)$$VA\left(\frac{d^{2}z}{dt^{2}}\right)=\sqrt{\frac{1}{T_{i}}\int_{0}^{T_{i}}{\left(\frac{d^{2}z}{dt^{2}}\right)}^{2}dt}$$
where *VA* is the power of the vibration signal for the observation period *T_i_*, which, for convenience, we received in one second. It is convenient to present the results of signal processing in the form of a table (see Table 1).

As can be seen from Table 1, along with the increase in the cutting speed, the power of the measured vibration signal also increases but, after reaching a speed of 140 m/min (2333 mm/s), the total vibration power is stabilized, due to the decrease in the power of the vibration signal in the direction of the *z* axis. The increase in vibration power with the increase in cutting speed is associated with the general vibration pattern of processing on an experimental machine; that is, the power increases due to the increase in the vibrations of the spindle group, as well as all moving parts of the machine. The drop in vibration power in the direction of the *z* axis is explained by the stabilization of the cutting force, which should be associated precisely with the regenerative effect, which, in turn, can be explained via the effect of the stabilization of the value of the actual feed in the direction of the *x* axis (see Expressions (4) and (5)). To increase the visibility when reducing the vibration power of the cutting tool, at the cutting speeds of 150 and 160 m/min, we present the dependence of the vibration acceleration along the z channel, in the form of a graph (see Figure 6).

As can be seen from Figure 6, the MCP curve has at least two local minima, each of which can be caused by the influence of the stability lobe of the regenerative effect. To clarify this hypothesis, we need to conduct a more detailed analysis of the data, in order to determine the values of the actual feed (see Expression (5)). The results of calculating the time constant, and the real feed value associated with it, for cases of cutting at the speed of 160 m/min, which corresponds to the minimum vibration acceleration that MCP removed from the tool in the direction of the *z* axis, are shown in Figure 7.

As can be seen from Figure 7, the value of the actual feed fluctuates significantly in the vicinity of the set value; we observe similar fluctuations in the calculated value of the delay time constant. For a more adequate analysis of the actual value of the feed, we will consider the calculated value of the vibration power of this value, for the calculation of which we somewhat transform Expression (6), removing the value of the average component of the feed from it:(7)$$VA\left(S\right)=\sqrt{\frac{1}{T_{i}}\int_{0}^{T_{i}}{\left(S-\bar{S}\right)}^{2}dt}$$
where $\bar{S}$ is a given constant value of the feed quantity.

The results of the calculation of the vibration power indicator of the actual feed value are presented in Table 2.

The analysis of the data presented in Table 2 shows that, at the cutting speeds of 150–160 m/min, a stabilization of the characteristic is observed, which indirectly confirms our hypothesis on the decrease in the influence of tool vibrations in the direction of giving. This decrease is due to the formation of such a real revolution period, in which the values of the current, and lagging via the amount of delay, and vibration displacement, compensate for each other. In other words, with some variation in the time delay, in Expression (5), the vibrations of the real feed are smoothed out.

To visualize the dependence of the actual feed value oscillation on the cutting speed, we present the graphical interpretation of the data given in Table 2 (see Figure 8).

As can be seen from Figure 8, the vibration power of the actual feed value actually has a declared local minimum in the vicinity of the same cutting speed value as in Figure 6. In other words, the area of stabilization of the cutting force in the direction of the *z* axis is due to the stabilization zone of the actual feed value.

Thus, the conducted studies have shown that, in addition to the vibration activity of the cutting tool in the feed direction, the regenerative effect depends on the vibration activity of the tool in the cutting direction, which affects the real value of the delay when cutting along the track.

## 5. Conclusions

The conducted studies have shown that, in addition to the vibrations of the cutting tool in the direction of feeding, the vibration activity of the tool in the direction of cutting plays an essential role in ensuring the regenerative effect. The delay time constant of the operator is formed, which determines the real value of the feed during cutting. The novelty of the research is that deformations in the direction of the cutting speed lead to the formation of a flexible positive feedback, which subsequently affects the dynamics of the cutting system. As this direction of tool deformation does not participate in the formation of the cut layer area and, as a consequence, in the formation of the cutting force, the effect of these fluctuations on the dynamics of the cutting system becomes unobvious. However, through changing the delay argument depending on the magnitude of these vibrations, the connection between the cutting force and the dynamics of the cutting process, and the vibrations of the cutting tool in the cutting direction, is revealed. It should be noted that the disclosure of this non-obvious connection affecting the properties of the cutting system complements modern ideas about the interconnectedness of the system of deformation movements of the tool, and the force reaction from the cutting process to the shaping movements of the tool, which determines the scientific novelty and significance of our research.

## Figures and Tables

**Figure 1 sensors-23-07482-f001:**
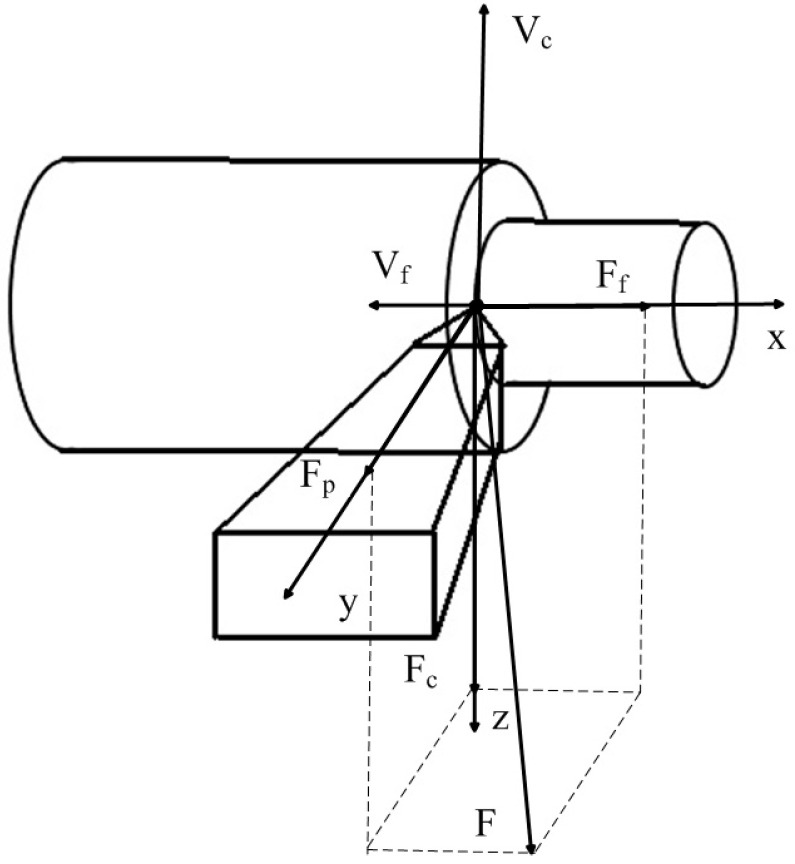
Diagram of the deformation vibrations and orientation of forces.

**Figure 2 sensors-23-07482-f002:**
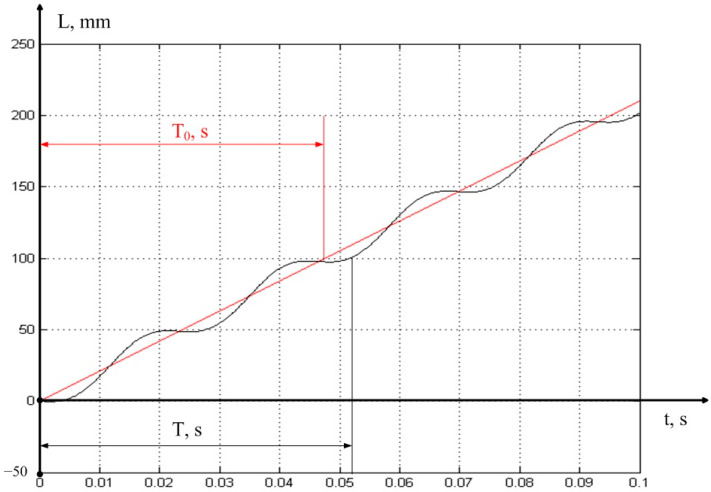
Example of the delay period variation during cutting.

**Figure 3 sensors-23-07482-f003:**
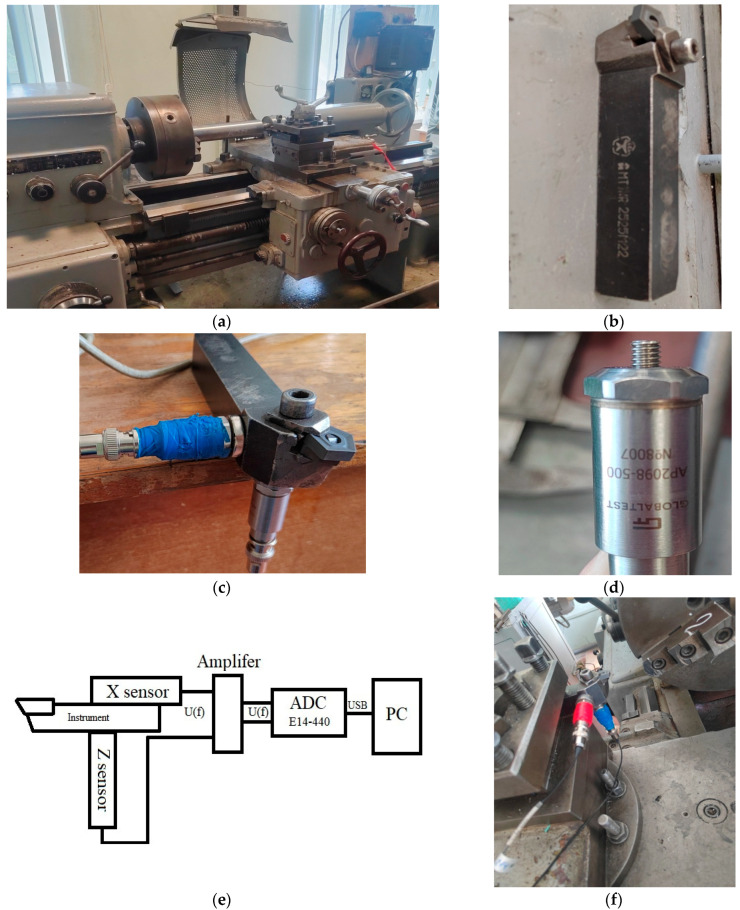
The measuring system: (**a**) the upgraded 1K625 machine, (**b**) the tool, (**c**) the tool with vibration converters AP2098-500, (**d**) the vibration converter AP2098-500, (**e**) the ADC and signal amplifier with a vibration converter, and (**f**) the measuring system on the machine.

**Figure 4 sensors-23-07482-f004:**
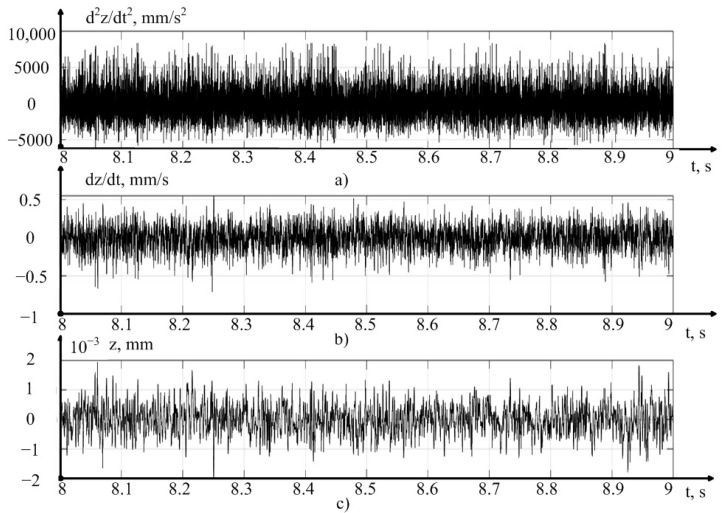
Experimental data for the case of the cutting speed of 140 m/min: (**a**) the measured signal of vibration acceleration in the direction of the *z* axis, (**b**) the average value of the vibration velocity in the direction of the *z* axis, obtained as a result of integration and subtraction, (**c**) the average value of the vibration displacement in the direction of the *z* axis, obtained as a result of double integration and subtraction.

**Figure 5 sensors-23-07482-f005:**
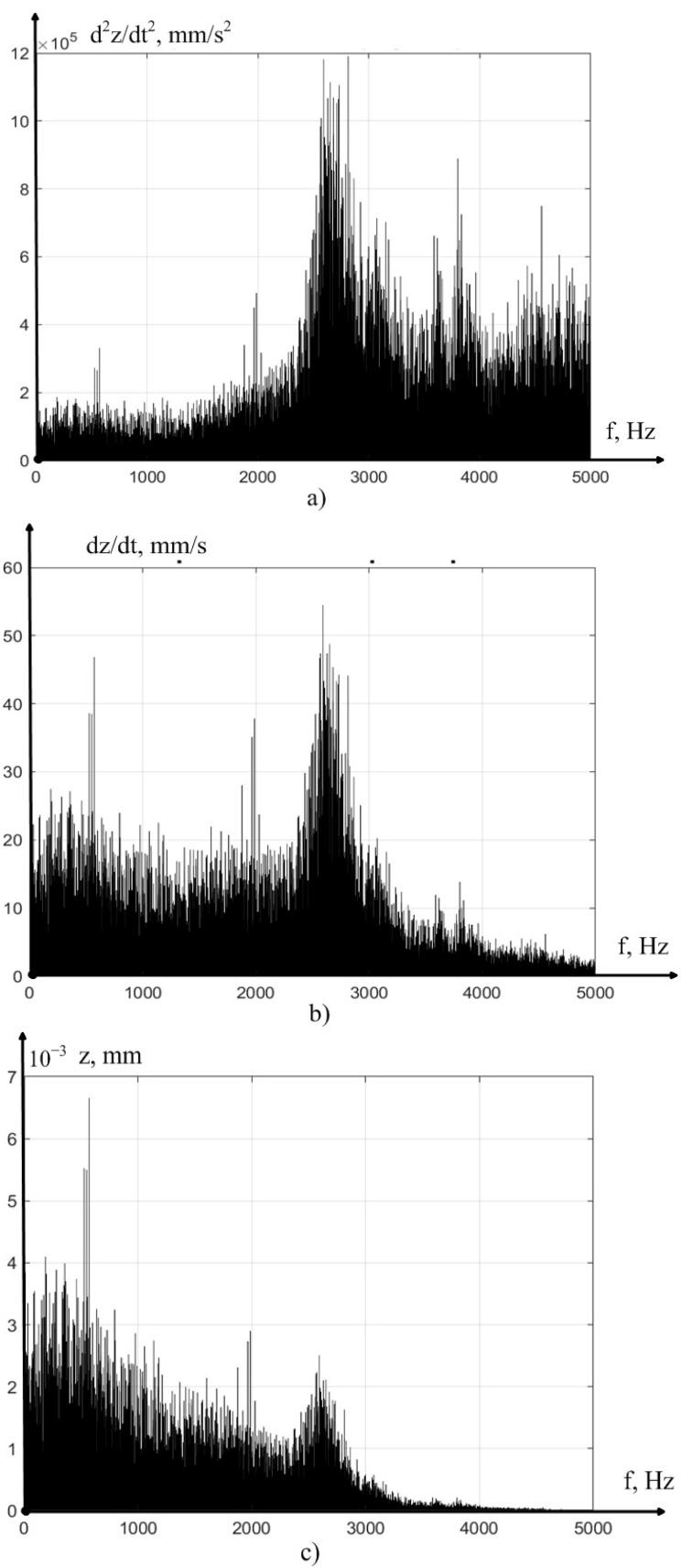
Signal power spectra: (**a**) the measured vibration acceleration signal in the direction of the *z* axis, (**b**) the vibration velocity in the direction of the *z* axis, (**c**) the vibration displacement in the direction of the *z* axis.

**Figure 6 sensors-23-07482-f006:**
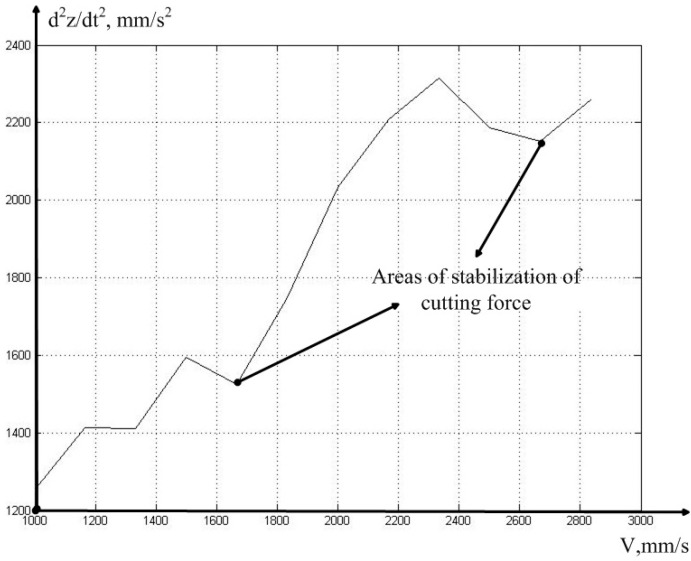
Areas of stabilization of the cutting force.

**Figure 7 sensors-23-07482-f007:**
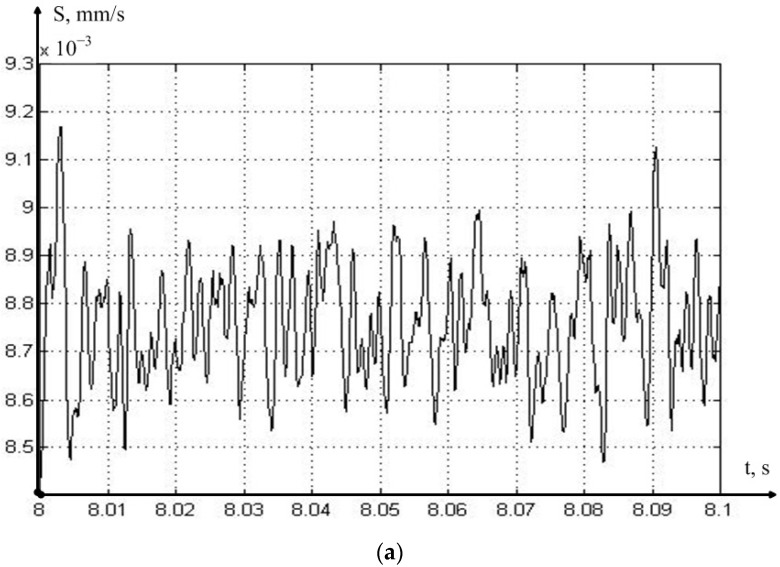
The value of the actual feed (**a**), and the corresponding value of the time constant (**b**) for the case of cutting at the speed of 160 m/min.

**Figure 8 sensors-23-07482-f008:**
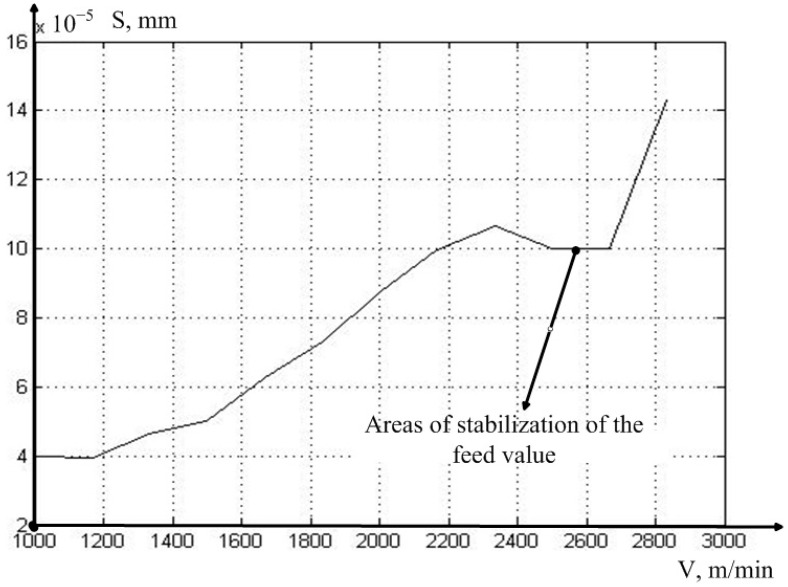
MCP of the feed value minus the set value.

**Table 1 sensors-23-07482-t001:** Results of the experimental data processing.

Cutting Speed, m/min	RMS Value of Vibration Accelerations		
	First Channel (x)	Second Channel (x) (z)	General Across the Two Channels
60	1026	1253	1145
70	1113	1414	1273
80	1132	1411	1279
90	1226	1596	1423
100	1455	1524	1490
110	1681	1748	1715
120	1883	2033	1959
130	2036	2207	2123
140	2268	2316	2292
150	2393	2187	2292
160	2505	2152	2335
170	2648	2260	2462

**Table 2 sensors-23-07482-t002:** MCP of the feed value minus the set value.

Cutting Speed, m/min	The RMS Value of the Feed Value Minus the Set Value
60	0.000039749
70	0.000039437
80	0.000046507
90	0.000050033
100	0.000062787
110	0.000072894
120	0.000087393
130	0.000099597
140	0.00010672
150	0.00010003
160	0.00010015
170	0.00014305

## Data Availability

Data on these studies will be published later in the master’s thesis Dudinov Ilia.

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
