# Peer review of "Research on Influence of Tool Deformation in the Direction of Cutting and Feeding on the Stabilization of Vibration Activity during Metal Processing Using Metal-Cutting Machines"

_sensors, 2023, doi:10.3390/s23177482_

Round 1

Reviewer 1 Report

This paper investigates the evaluation of the impact of the change in feed direction delay value on the machining dynamics of metal-cutting machine tools, which is the determining factor of the self-excitation regeneration characteristics of metal-cutting systems. The surveys in this paper are sufficient, and the topic is relatively practical. But the differences and advantages compared with other research work are not demonstrated enough. Comments and suggestions are listed as follows.

1. In the introduction, the novelty of this article is not fully reflected. There is a lack of discussion on other experimental methods. It is suggested to add the recent experimental progress between the third and fourth paragraphs of the introduction.

2. Some figures are unclear. Figure 4 suggests using screenshots instead of taking photos.

3. Figures 1, 7, 8, and 9 are too large to view.

4. It is recommended that Figure 6 (a) is the general diagram of the experimental device, and other modules should be marked in this figure.

5. The physical symbols in the text and formulas should be unified in italics. In addition, the extra space before "z axis" affects reading.

6. During the machining process, there is a periodic oscillation of a force, which is related to both material properties and machining parameters. It is recommended to further consider the characteristics of this force in the author's work.

doi.org/10.1016/j.jmapro.2021.04.075

Minor editing of English language required

Author Response

Thank you for your work in assessing the quality and scientific novelty of our article. We have taken into account all your recommendations. We have made appropriate changes to the materials of the article.
1. The introduction has been changed accordingly, we have considered new and promising methods for constructing measuring systems. Here we also discussed the disadvantages of these methods when measuring tool vibrations in metalworking.
2. Figure 4 was removed from the work, both on your comment and on the comment of the second reviewer. The drawing is really low-informative and of poor quality.
3. We have reduced the corresponding figures.
4. We have difficulties with this remark. However, after discussion, we came to the conclusion that we are talking about 3, not 6 figure. In this regard, we have made appropriate changes to Figure 6. We hope that we have understood you correctly.
5. We have made appropriate corrections to the text of the work in accordance with your comment. Note that this is our mistake. Apparently we looked at it earlier.
6. With regard to this point of comments, we have added to the article a mention of the fact that the cause of vibrations of the cutting force may be the internal molecular features of the processed metals. However, in our work, the emphasis is placed on secondary, internal in terms of cutting force, causes of vibrational activity. Therefore, the entire measuring system is built around this mechanism of vibration activity in the cutting system.
Thank you again for your attention to our work, we hope that we have responded correctly to your comments.

Reviewer 2 Report

Dear authors,

Below please find some remarks considering your paper.

·         In the case of the title and the abstract, those elements must be understandable to none familiar with the field reader. Some sentences in the abstract are over complicated and thus difficult to understand. Additionally, I would suggest putting some emphasis on the novelty of the topic in the abstract.

·         The measurement setup- not everything is clear from Fig.3. As this is a contact way to measure vibration is there an impact on system dynamics from the way the sensors were attached to the tool? Was It evaluated in any way or cross-checked wit another none-contact method?

·         Talking about the methods used they were not introduced and no state-of-the-art in case of measurement techniques was introduced in the experimental chapter or/especially in the introduction. However, there are numerous techniques, especially none contact ones that are especially profitable for the evaluation of such cases. Some examples include video motion amplification (https://doi.org/10.1016/j.measurement.2022.112218), using a microphone for obtaining FRFs or especially using Laser Doppler Vibrometer. This last one is especially interesting in the case of blade evaluation (e.g. DOI: 10.3390/s23031263.). These alternative ways, together with the methods used by authors, should be evaluated as a part of state-of-the-art analysis. In the reviewers' opinion, the introduction is rushed and incomplete.

·         This last comment is also connected with the reference and the introduction where it looks like the authors focused on the research performed in the past in Russia and on research from Russian researchers. It is noticed that this region is famous for this kind of research but the reviewer urges the authors to look at the case of state-of-the-art analysis wider.

·         No information can be obtained from Fig.4. This does not look professional. This figure must be improved.

·         Fig.6 is very important. If possible please check if maybe the presentation of this data in logarithmic scale is not giving a better sense of frequencies associated with the highest amplitudes.

·         The conclusion and discussion are good.

In conclusion: The paper is clear. The biggest advantage is the presentation of tests in real-life applications. The results presentation is good however it would profit from the improvement of graphical outputs. The paper needs however some editing and especially improvement in the introduction section.

Hope the authors will use some suggestions to improve this otherwise very interesting paper.  

Author Response

Thank you for your work in assessing the quality and scientific novelty of our article. We have taken into account all your recommendations. We have made appropriate changes to the materials of the article.
1. We have completely rewritten the annotation to the article, where we indicated aspects related to the novelty of our work.
2. In the introduction, we considered the reasons for building a measuring system on this contact method of measurement. We did not duplicate measurements using other methods. So our proposed method is the most reliable, in our opinion. However, we are aware that the measuring system can be improved. Here we see the introduction of a three-stage vibration accelerometer into the tool holder, the output of which will be isolated from the impact of aggressive factors arising during the cutting process. Such a tool will solve many research problems, as well as provide optimal processing modes. We see the development of such a system as a logical continuation of this work.
3 and 4 points of comments. We have tried to make appropriate corrections to the materials of the introduction, including expanding the list of sources used by us. As for the fact that we use a large number of references to publications of Russian scientists, this is probably due to the influence of our scientific school. The point is that the research presented in the article is organically connected with previous research in this area conducted by both our research team and the teams with which we interact directly. We hope that publishing our research in the world's leading scientific publications will allow us to increase our level of communication in this area.
5. We agree with the comments on Figure 4, in this regard, we decided to remove it from the materials of the article. We believe that it is enough to leave a description of the ADC software complex. This will be enough for specialists in this field.
6. We decided not to present Figure 6 (in the new numbering 5) on a logarithmic scale. This is due to the fact that we did not see a significant change in the spectra shown in this figure.
Thank you again for your attention to our work, we hope that we have responded correctly to your comments.

Round 2

Reviewer 1 Report

Accepted